# Molecular Mechanisms of Kaposi Sarcoma Development

**DOI:** 10.3390/cancers14081869

**Published:** 2022-04-07

**Authors:** Andy Karabajakian, Isabelle Ray-Coquard, Jean-Yves Blay

**Affiliations:** 1Department of Medical Oncology, Centre Léon Bérard, 69008 Lyon, France; andy.karabajakian@lyon.unicancer.fr (A.K.); isabelle.ray-coquard@lyon.unicancer.fr (I.R.-C.); 2Faculty Medicine Lyon Est, University Claude Bernard Lyon I, INSERM 1052, CNRS 5286, Centre Léon Bérard, 69008 Lyon, France; 3Department Teri, Centre de Recherche en Cancérologie de Lyon, 69008 Lyon, France

**Keywords:** Kaposi sarcoma, KSHV, virus, sarcoma, immunosuppression, growth factors, oncogenesis

## Abstract

**Simple Summary:**

There are at least four forms of Kaposi’s sarcoma (KS) with the ‘HIV’-related form being the most aggressive and can involve mucosae or visceral organs. Kaposi’s sarcoma-associated herpes virus (KSHV) is the underlying cause of this disease. It can infect endothelial and/or mesenchymal cells and establish a latent phase in host cells in which latency proteins and various non-coding RNAs (ncRNAs) play a complex role in proliferation and angiogenesis. It also undergoes periods of sporadic lytic reactivation that are key for KS progression. Complex interactions with the microenvironment with production of inflammatory cytokines and paracrine signaling is a standout feature of KS development and maintenance. KSHV impairs the immune response by various mechanisms such as the degradation of a variety of proteins involved in immune response or binding to cellular chemokines. Treatment options include classical chemotherapy, but other novel therapies are being investigated.

**Abstract:**

Kaposi’s sarcoma (KS) is a heterogeneous angioproliferative tumor that generally arises in the skin. At least four forms of this disease have been described, with the ‘HIV’-related form being the most aggressive and can involve mucosae or visceral organs. Three quarters of KS cases occur in sub-Saharan Africa (SSA) as geographic variation is explained by the disparate prevalence of KS-associated herpes virus (KSHV), which is the underlying cause of this disease. It can infect endothelial and/or mesenchymal cells that consequently transdifferentiate to an intermediate state. KSHV establishes a latent phase in host cells in which latency proteins and various non-coding RNAs (ncRNAs) play a complex role in proliferation and angiogenesis. It also undergoes periods of sporadic lytic reactivation triggered by various biological signals in which lytic stage proteins modulate host cell signaling pathways and are key in KS progression. Complex interactions with the microenvironment with production of inflammatory cytokines with paracrine signaling is a standout feature of KS development and maintenance. KSHV impairs the immune response by various mechanisms such as the degradation of a variety of proteins involved in immune response or binding to cellular chemokines. Treatment options include classical chemotherapy, but other novel therapies are being investigated.

## 1. Kaposi’s Sarcoma: Definition & Epidemiology

In 1872, Moritz Kaposi, a Hungarian physician, first noted multifocal pigmented skin lesions in elderly European and Mediterranean men. He was describing what is now known as the “classic” form of Kaposi’s sarcoma (KS) [1], which is typically an indolent disease, and primarily affects skin on the legs. In the middle of the 20th century, cases of “endemic” lymphadenopathic forms of KS were being described in children and young/adults of sub-Saharan Africa (SSA) [2]. Later in the 1960s, an “iatrogenic” form of KS was observed in patients receiving immunosuppressive medication and organ transplant recipients [3]. Incidence of “iatrogenic” KS is reported to be 200-fold higher in solid-organ transplant recipients than in the general population [4]. More recently in the beginning of the 1980s, the “epidemic” or acquired immunodeficiency syndrome (AIDS)-related KS was described by the Centers for Disease Control (CDC) at the onset of the AIDS epidemic affecting homosexual men [5,6]. A fifth form that occurs at any skin site, usually indolent with few lesions, is hypothesized to exist in young or middle-aged homosexual men without HIV infection [7]. All epidemiological forms of this angioproliferative tumor generally arise in the skin of the extremities as nodular lesions, multiple plaques, or patches, and less frequently on mucosa and visceral sites mostly in immunosuppressed patients.

The “epidemic” HIV-related form is particularly aggressive and can involve mucosae or visceral organs [8,9]. Compared with the general population, KS rates are elevated 500-fold among people living with HIV (PLWH) [10], but epidemiologists were convinced for many years that KS was caused by a second oncogenic virus [11]. In 1994, a breakthrough discovery of the human herpesvirus 8 (HHV8), also known as KSHV (Kaposi’s sarcoma-associated herpes virus), was identified in KS lesions obtained from AIDS patients [12].

The incidence of KS varies considerably around the world [13,14]. According to the latest GLOBOCAN estimates, nearly three-quarters of global KS cases in 2020 occurred in sub-Saharan Africa (SSA) with around 34,270 cases diagnosed worldwide [13,14]. Before the AIDS epidemic, it was a rare disease (0.01 to 1 per 100,000/year in Europe) affecting men more frequently than women. The incidence is reported to be >200-fold higher in immunocompromised populations (AIDS, organ transplantation). Since the emergence of the AIDS epidemic in the 1980s, estimating the incidence of “non-HIV related” KS has become more difficult. After a dramatic increase in the early phase of the epidemic [15], the “HIV-related” KS incidence decreased significantly with the introduction of combination antiretroviral therapy (cART) in PLWH, especially in high-income settings [16]. In a recent epidemiological study, it was estimated that around 70% of the current global KS burden could be prevented by eradication of HIV infection [17]. In 2022, on the IARC website [13], the incidence is reported to be 3.7/10^6^/year. In a recent epidemiological nationwide study conducted in France, the incidence of KS was 2.5/10^6^/year, with 18% occurring in an HIV context, and 28% in patients with immunodepression [18]. In another study conducted in three EU regions, the incidence was found to be 3.7/10^6^/year, higher in the Italian region (Veneto) as compared to two other French regions (Auvergne-Rhone-Alpes, Aquitaine) [19].

## 2. KSHV Incidence

This geographic variation of KS is driven by the disparate prevalence of KSHV, which is the underlying cause of all forms of KS, and HIV infection that worsens the carcinogenic outcome of KSHV infection [14]. The prevalence of KSHV is lowest in the United States, Asia, and Northern Europe, where it affects less than 10% of the population, higher in some areas in the Mediterranean (20–30%), and highest in regions of equatorial SSA where it can reach more than 90% [14]. Saliva and close sexual contact are the main transmission routes, and to a lesser extent, through blood [20,21]. Regardless of the region or population group, a meta-analysis showed that HIV infection is associated with a high prevalence of KSHV, with a strong association for men that have sex with men (MSM) and weaker for heterosexual adults and intravenous drug users [22]. Several mechanisms might explain this association, including reactivation of pre-existing KSHV in PLWH, increased susceptibility to KSHV infections in PLWH, and shared transmission routes [23].

## 3. Kaposi’s Sarcoma-Associated Herpes Virus (KSHV)

KSHV, also known HHV8, is a double-stranded linear DNA gamma-herpesvirus and is the causal agent of all forms of Kaposi sarcomas. It is also the underlying cause of primary effusion lymphoma (PEL) and multicentric Castleman Disease (MCD) [14]. A link with osteosarcoma has also been recently suggested in an epidemiological study of the Uyghur population [24]. KSHV contains a double-stranded DNA with 90 open reading frames, an icosahedral capsid, a tegument, and an envelope [25]. It can infect a variety of cell types using viral envelope glycoproteins, including monocytes, dendritic cells, fibroblasts, B cells, and endothelial cells [26]. Receptors of KSHV include heparan sulfate binding proteins; α3β1, αVβ3, ανδ αVβ5 integrins; EphA2 receptors; and DC-SIGN [27].

Similar to other herpesviruses, KSHV establishes latency in host cells (endothelial cells, mesenchymal cells, and B cells in this case), responsible for lifelong infection. During latency, only a few genes from the latency locus are expressed, such as open reading frame (ORF)71 (v-FLIP), ORF72 (v-Cyclin), ORF73 (latency-associated nuclear antigen, or LANA), and the K12 region (which gives rise to at least 12 viral miRNAs) [28]. LANA binds to histones and enables the tethering of the latent KSHV circular episome to the host chromosome so that it can be replicated during cell division [29]. Latent gene-encoded proteins ensure cell survival by several different mechanisms like the inhibition of apoptosis or by stimulation of the nuclear factor-κB (NF-κB) pathway [30,31]. Latency is the default pathway of the virus, but KSHV undergoes sporadic periods of lytic reactivation [26,31]. During the lytic replication phase, all genes are expressed in a temporal order (immediate-early genes (IE), delayed-early genes (DE), and late genes (LG) so that the viral genome is replicated. The infectious virions are then released via budding and death of the host cell, enabling the infection of other cells. Paradoxically, lytic reactivation is an important feature of KS progression and oncogenesis, as discussed hereunder. Protein products of the IE phase control transcription and one of these proteins; replication and transcription activator (RTA) is the master switch transcription factor ensuring the activation of viral and cellular promoters [25,31]. The DE phase proteins are responsible for viral DNA replication, and the late phase ensures the expression of viral structural proteins and the production of the infectious virus [25,31].

## 4. Histology and Cellular Origin of KS

KSHV affects endothelial cells and/or mesenchymal cells that consequently adopt an endothelial differentiation. Early lesions (plaque or patch) often appear as a granulation type reaction with immune cell infiltration, intense angiogenesis, and proliferating “spindle”-shaped cells of endothelial and macrophagic cell origin, which are the tumor cells of KS. A typical KS is characterized by proliferation of spindle-shaped endothelial cells in the skin and sometimes in mucosal and visceral sites. Kaposi cells form slit-like vascular spaces, with often extravasated inflammatory cells and red blood cells. In late, nodular, lesions of KS, tumor cells eventually become the predominant cell type and lesions acquire a fibrosarcoma-like aspect, though neoangiogenesis is present [9]. KS cells express both pan-endothelial and lymphatic endothelial markers (e.g., ERG) and may also appear as poorly differentiated spindle cells expressing markers of a variety of cell types. KSHV infection causes reprogramming of the cellular gene expression, as well as a morphological change of the cells to KS “spindle cells” [31,32]. KS cells express the LANA protein of KSHV, which is detectable on IHC and is used to confirm the presence of KSHV in a suspected KS lesion. As mentioned above, LANA is a viral protein essential for viral episomal persistence. The infected endothelial cell then undergoes an endothelial to mesenchymal transition, a phenomenon possibly related to the capacity of the LANA to induce mesenchymal cell programs [30]. KSHV has also been reported to infect mesenchymal stem cells [33,34]. A process of transdifferentiation to an intermediate state of both endothelial and mesenchymal cells either by mesenchymal-to-endothelial transition or by endothelial-to-mesenchymal transition occurs in response to viral infection [33,34]. Mesenchymal stem cells infected with KSHV also express a set of makers of both mesenchymal and endothelial cells and produce PDGF and other angiogenic factors such as VEGF & FGF [33,34].

Spontaneous reactivation of KSHV in lymphatic endothelial cells is important to maintain and expand the population of infected cells in the tumor and to promote tumor cell expansion. The mechanisms of activation of the lytic stage in these cells may involve specific transcription factors. Prox1 is a lymphatic endothelial cell-specific transcription factor that induces the expression of the RTA gene to promote lytic cycles, while SOX18, another transcription factor necessary for lymphatic endothelial cell development, promotes an increase in the number of latent genomes. Both are induced by KSHV infection and play an important role in KS oncogenesis, Prox1 being necessary to maintain the number of infected cells in the tumor [35,36].

## 5. Molecular Mechanisms of Transformation

Oncogenic DNA viruses produce oncogenic proteins responsible for cellular proliferation and the protection of the host cell against apoptosis. They evade the host immune surveillance system by multiple mechanisms such as the reduction of viral gene expression to limit the amount of viral antigen presentation, the synthesis of critical proteins affecting the process of antigen presentation, and the expression of various viral “immunoevasins” that manipulate host cells to reduce immunogenicity [37,38] (Table 1).

Following KSHV infection of endothelial cells (or mesenchymal stem cells), it establishes a latent phase, expressing a limited set of genes, including LANA (ORF73), vFLIP (ORF71), vCyclin (ORF72), Kaposins (ORFK12), and a set of miRNAs. Some of these proteins are analogs of cellular genes (FLIP, Cyclin D) and have oncogenic properties in vitro in cell cultures [37,38].

These latent proteins, along with the miRNAs, play a complex role in cell survival, proliferation, and angiogenesis. They promote the tumorigenesis of KS. ORF72 (vCyclin) expression overcomes senescence in infected endothelial cells and activates a telomere maintenance mechanism essential for the oncogenic process in KSHV-infected cells [39,40]. LANA induces the cleavage of the Aurora B kinase and increases expression of the pro-survival factor MCL-1 [41,42]. KSHV encodes various non-coding RNAs (ncRNAs) that are also found in humans, such as microRNAs (miRNAs), small and long non-coding RNAs (lncRNAs), and circular RNAs (circRNAs) [43,44]. These KSHV ncRNAs play an important role in transformation [43]. Some miRNAs block apoptosis by targeting the pro-apoptotic CASP3 gene [45] or the mTOR inhibitory factor CASTOR1 [46]. KSHV miRNAs also target immune-regulatory gene transcripts in a collaborative way in order to evade immune surveillance [43]. miR-K12-9 targets MYD88 and IRAK1 transcripts thus impairing the Toll-like receptor (TLR) signaling cascade. miR-K12-5 downregulates MyD88, impairing downstream pro-inflammatory signals [47]. CircRNAs are a novel class of ncRNAs playing a role in tumorigenesis of KS by promoting cell proliferation [48].

## 6. Lytic Cycle and KS Progression

Lytic cycles of KSHV are observed in lymphatic and endothelial cells and in KS cells. They play a key role in the progression of KS in the clinic (Table 1). The induction of lytic genes requires the sequential expression of IE genes, DE genes, and LG. IE gene expression do not require protein synthesis, indicating that cellular proteins, epigenetic regulators, and host transcription factors are involved in KSHV reactivation. The activation of the lytic stages is triggered by a variety of biological signals. It involves the modulation of cellular epigenetic machinery, the activation of signaling pathways (MAPK), the modulation of viral and host ncRNAs, and a defective function of the immune system [31]. All these events converge to counteract LANA-mediated repression of lytic gene transcriptionand the induction of RTA, which transactivates KSHV lytic genes [31].

Several lytic stage proteins modulate host cell signaling pathways to allow cell survival and proliferation and to augment viral replication [31]. In the clinic, the progression of KS requires the development of lytic cycles in vivo to disseminate the virus within the host and maintain the pool of latently infected cells [14]. Though paradoxical, this was suspected early with the observation that anti-CMV agents were able to induce KS responses in AIDS patients. Antibodies against lytic proteins can be detected in patients that will develop KS prior to the onset of disease, suggesting that high levels of reactivation from latency occur before disease development [49]. In vitro, KSHV is rapidly lost in cell cultures. Thus, the persistence of a pool of latently infected cells in vivo requires the activation lytic cycles. This consequently results in the production of oncogenic proteins, as well as secreted proangiogenic and pro-inflammatory cytokines essential for the oncogenesis of KS [50,51,52,53,54,55]. The activation of lytic cycles is also associated with the production of viral proteins that play a key role in KS tumor progression and are detectable in KS lesions. These include viral oncogenes and intracellular or membrane proteins, such as vGPCR, which is an analog of IL-8R (not requiring the presence of the ligand to deliver the signal).vGPCR induces the production of cytokines such as PDGF and VEGF, which is considered important for the development of the lesions. Transgenic mice expressing vGPCR develop KS-like lesions [50]. Other proteins are produced during the lytic cycle such as vIL-6, detectable in the peripheral blood of KS patients [14], and viral chemokines, which play an important role in KS growth and progression by modulating immune response. The K1 and K15 membrane proteins also exert oncogenic properties. K1 is expressed at low levels in latency and higher during the lytic cycle; K1 has an ITAM signal in its intracellular domain, enabling signal transduction. K1 activates the PI3K/mTOR pathway, blocks Fas-mediated killing, induces VEGF, and activates the MAPK pathway. The viral protein kinase (vPK) is a ribosome S6 kinase analog, a cellular protein acting downstream of mTOR [51]. K15 is also a transmembrane protein that expressed a low level in latency at higher levels in the lytic cycle, which activates the MAPK pathway as well as NFkB. Both transmembrane proteins are detectable in KS lesions [52].

The oncogenic proteins of KSHV thus activate a series of cellular pathways, the MAPK, NFkB, and PI3K pathways, as well as the production of proinflammatory cytokines and angiogenic factors, which play a key role in the oncogenesis of KS. One of the intriguing features of KSHV-induced oncogenesis is the contribution of soluble factors.

## 7. Cytokines and Paracrine Growth Factor Production in KS

The infection of endothelial cells with KSHV results in cell transformation, accompanied by complex interactions with the microenvironment (Table 1). Infected cells produce cytokines that induce paracrine signaling, including vascular endothelial growth (VEGF), PDGF, FGF, and interleukin 6 (IL-6). These cytokines are implicated in the development and maintenance of KS [43]. Murine mesenchymal stem cells are not transformed by KSHV infection, but require coculture in KS-derived media to acquire a transformed phenotype [55].

The previously mentioned lytic gene ORF74, encoding for the constitutively active vGPCR, activates MAP kinase and NFkB pathways [55], inducing the production of VEGF, IL-6, IL-8, and tumor necrosis factor alpha, and promoting tumorigenesis through autocrine and paracrine pathways [56,57,58,59,60]. Other lytic genes including vPK, viral IL-6 homolog (vIL-6/K2), viral interferon regulatory factor 1 (vIRF1/K9), and the membrane proteins K1 and K15 have oncogenic functions [56,59,60]. KSHV vIL-6 can activate the JAK–STAT, PI3K-AKT, and MAPK–ERK pathways upon binding directly to gp130 dimers inducing the production of cellular IL-6, along with other viral proteins (Kaposins) [56,57,58]. It was shown more recently that vIL-6 decreases the expression of caveolin 1 and increases the expression of integrin β3 through its activation of STAT3, which contributes to its angiogenic-like behavior [59,60]. ORF36 phosphorylates ribosomal protein S6 to increase protein synthesis and growth, angiogenesis, and cell proliferation [59].

All these elements are responsible for the production of inflammatory cytokines, with paracrine and endocrine paraneoplastic activities.

## 8. Immune Evasion

The immune system plays an important role in the control of KS (Table 1). Restoration of immune system function with cART in AIDS patients with progressive KS is often sufficient to induce tumor response in HIV-related KS [14]. There are numerous mechanisms by which KSHV impairs the immune response.

KSHV produces proteins and miRNAs that impair cytotoxic T or NK cell-mediated killing [60]. KSHV K3 and K5, both ubiquitin ligases, promote the degradation of a variety of proteins involved in immune response, including MHC-I, CD1d, CD31, IFN-gR1, CD54, B7-2, CD1d, MICA, and MICB, interfering with CD8+ T cell and NK cell function [62,63,64,65]. KSHV miR-K12-7 downregulates MICB expression [63]. vOX2 protein, a cellular ortholog of CD200 expressed in the lytic phase, was found in-vitro to suppress antigen(Ag)-specific T cell response. KSHV protein RTA also impairs CD4+ T cell function, promoting the degradation of MHC-II and inducing MARCH-8, a MHC-II antagonist [66].

The inhibition of immune response induced by KSHV occurs at multiple steps of the development of the immune response, from the initial activation of innate immunity. KSHV viral interferon regulatory factors (vIRFs) impair TLR signaling at multiple levels: vIRF reduce TLR3-mediated interferon induction [67]. The RTA protein also regulates TLR signaling by promoting proteasomal degradation of the downstream TLR3 adaptor protein TRIF, as well as downregulation of TLR2 [68,69]. RTA also blocks TLR4 signaling by promoting degradation of MyD88 mRNA [70]. KSHV infection in endothelial cells also rapidly suppresses TLR4 signaling via vGPCR and vIRF1-mediated mechanisms [71].

KSHV induces the production of viral cytokines and chemokines while inducing the production of cellular inflammatory cytokines. This is observed in vitro when monocytes are infected by KSHV, and in vivo in patients with KS. These increased serum levels of IL-6, IL-10, and TNF, and detectable vIL-6, are indeed observed in patients with progressive KS [62]. These cytokines are known to exert proliferative activities directly on tumor cells, but also immunosuppressive functions (IL-6, IL-10) [72,73], as well as paraneoplastic inflammatory activities, referred to as KSHV-associated inflammatory cytokine syndrome (KICS), with very high circulating levels of IL-6 and IL-10 [74,75,76]. KSHV also modulates IL-4/13-STAT6 signaling by inducing constitutive phosphorylation of STAT6 through a LANA-mediated cleavage and subsequent nuclear translocation of STAT6 contributing to evasion from immune surveillance [77,78].

KSHV also contains three viral homologs of cellular chemokines, vCCL1 to vCCL3 (also known as vMIP-I to vMIP-III). vCCL2 binds to a wide variety of chemokine receptors on many different cell types and can have agonistic or inhibitory effects. vCCL1 binds to CCR8, which is prominently expressed on Treg cells, suggesting that the virus may recruit this T cell subtype [79]. vCCL2 has been shown to bind CX3C chemokine receptor 1 (CX3CR1) and CCR5, blocking their natural ligands and inhibiting the migration of naive and activated NK cells [79]. vCCL2 also antagonizes CCR1 and CCR5 activation, which are primarily found on Th1 cells, while stimulating CCR3 and CCR8, which has been shown to attract Th2 cells to KS lesions [79]. By promoting Th2 cell attraction, this protein may allow immune escape of infected cells. vCCL3 is an agonist of CCR4, another chemokine receptor found preferentially on Th2 and Treg cells. vCCL3 also preferentially induces chemotaxis in Th2 cells when compared to Th1 cells, suggesting that viral chemokines may play a role in creating the characteristically Th2-skewed KS microenvironment [79].

## 9. Genetic Predispositions to Develop KS after KSHV Infection

As mentioned previously, there is a high individual variability in disease development and presentation within all forms of KS. Genetic differences between populations could explain this heterogeneity, suggested by the observation of familial clustering of KS cases [80]. Polymorphisms in genes of the HLA subtype, in immune regulation genes such as NFκB or LY6G6C, and in subtypes of the NK cell receptor KIR, have been linked to KSHV infection or development of KS [81]. It is not yet clear how and if these polymorphisms modify the course of the disease.

## 10. Therapeutic Implications

To what extent can the partial understanding of the oncogenesis of KS guide us to new treatment decisions?

There are multiple active treatments of KS. While classic chemotherapy has been used for decades, many clinical trials evaluating immune checkpoint inhibitors or anti-angiogenic agents are underway in both HIV and non-HIV-related KS. The treatment of KS in the clinic is based on consideration of multiple criteria including patient’s comorbidities and preferences, disease extension, and the immunological and virological status of the patient [82]. Since KHSV cannot be eradicated from the body, KS cannot be cured, but long term remissions are now routinely observed in particular when an underlying immunodepression has been corrected. Modification of immunosuppressive therapies for iatrogenic-KS or optimal control of HIV infection with cART for HIV-associated KS should be the first preferred option whenever possible [83].

If insufficient, e.g., in advanced or very symptomatic disease, chemotherapy can be considered. Pegylated liposomal doxorubicin (PLD) is a standard first line [84]. Paclitaxel is also commonly prescribed in first-line and can be safely used with cART [85].

Other front-line options include docetaxel and nab-paclitaxel as small studies have shown comparable efficacy and acceptable toxicity [86,87]. Oral Etoposide, gemcitabine, bleomycine, and vinca-alcaloids, are all options after failure of first-line chemotherapy [88,89,90,91]. However, these are not specific targeted therapies.

With the identification of the role of endogenous cytokines, angiogenic factors, and immune suppressive factors in KS progression, anti-angiogenic agents & immunomodulators are investigated [92,93,94,95,96,97,98,99]. The monoclonal anti-VEGF antibody bevacizumab resulted in a 31% response rate (RR) in monotherapy and up to 56% in combination with PLD [94,95]. Tyrosine kinase inhibitor sorafenib is a potent antiangiogenic, but activity was found modest in an early phase trial with a 29% RR [96].

Pomalidomide is an immunomodulatory agent capable of reversing viral-induced downregulation of immune surface receptors, rendering KS cells more immunogenic. Promising results were observed in prospective trials with RRs up to 40% for HIV-related KS [92,93]. Chronic T-cell stimulation by HIV and oncogenic viruses can lead to an increase in immune checkpoint receptors on T cells, leading to an eventual T-cell exhaustion [97]. Nevertheless, PLWH were excluded from the majority of the clinical trials testing immune checkpoint inhibitors, including the anti-PD-1/PDL-1 antibodies that have now revolutionized cancer care. The main concerns were the reactivation of HIV and toxicity [98]. In a recent phase Ib trial, monotherapy with pembrolizumab in HIV-related KS resulted in a stable disease (mostly tumor regression not meeting the criteria of partial response) in five out six patients [99], but reports emerged on KSHV lymphoproliferation potentially attributed to pembrolizumab.

KICS is a severe life-threatening condition related to an overwhelming production of proinflammatory cytokines [76]. Treatment with anti-cytokine antibodies such as anti-IL-6 was reported as active in a small series of single cases of KS patients [76,77,78], consistent with that observed for other sarcomas [100]. Other early-phase trials are currently ongoing, including combination therapies such as cabozantinib and nivolumab [NCT04514484], pomalidomide and nivolumab [NCT04902443], and ipilimumab and nivolumab [NCT02408861]. Finally, the better understanding of the molecular mechanisms of KS oncogenesis, including the identification of key viral protein targets offers new avenues for innovative therapies. Activating the unfolded protein response (UPR) by the glucose analog 2-deoxy-d-glucose (2-DG) was shown to elicit an early antiviral response via eukaryotic initiation factor 2 (eIF2α) inactivation, which impairs protein synthesis required to drive viral replication and oncogenesis. Thus, induction of endoplasmic reticulum (ER) stress by 2-DG might be a promising strategy [101]. Potent in-vitro inhibitory activity against KSHV replication by the protease inhibitor (PI) nelfinavir was demonstrated in another robust preclinical study [102].

Targeting viral miRNAs has promising potential as viral miRNAs are distinct from cellular miRNAs. In a recent study, targeting three KSHV miRNAs with antisense inhibitors suppressed the growth of KSHV lymphoma cells in mouse xenograft models [103].

## 11. Conclusions

The understanding of the mechanisms of cellular transformation due to KSHV in KS has considerably progressed 28 years after the identification of the virus. Multiple steps and viral oncoproteins are involved in the transformation process, which requires a complex interplay between the viral oncoproteins and cellular proteins as well as the modulation of the host’s immune system response. This opens new avenues for the treatment of this multifaceted disease that affects large human communities across the globe.

## Figures and Tables

**Table 1 cancers-14-01869-t001:** An overview of the mechanisms of Kaposi Sarcoma development.

Main Steps	Main Elements	Oncogenic Roles	Mechanisms	Refs
Latent phase proteins and RNAs	LANA (ORF73), vFLIP (ORF71), vCyclin (ORF72), Kaposins (ORFK12), Viral miRNAs	Cell survival, proliferation, angiogenesis	Telomere maintenance Increased expression of pro-survival factors RNA inhibition of apoptosis or immune evasion by targeting various gene transcripts	[41,42,43,44,45,46,47,48]
Lytic activation	Immediate-early (IE) genes Delayed-early (DE) genes & proteins Late genes & proteins (K1, K15)	Cell survival & proliferation, viral replication	Production of proangiogenic (PDGF, VEGF) and pro-inflammatory cytokines (vIL-6), viral oncogenes (vGPCR) MAPK, PI3K, and NFkB pathway activation	[49,50,51,52,53,54]
Soluble growth factors	Cytokines: VEGF, PDGF, FGF, IL-6, IL-8, tumor necrosis factor alpha, vIL-6 Chemokines	Development & maintenance Protein synthesis & growth	Autocrine & paracrine pathway signaling Activation of JAK–STAT, PI3K-AKT, and MAPK–ERK pathways	[55,56,57,58,59]
Immune evasion	Viral proteins: K3, K5, vOX2, RTA, viral interferon regulatory factors (vIRFs) Viral miRNAs: miR-K12-7 Cytokines: IL6, IL10 Viral chemokines: vCCL1, vCCL2, vCCL3	Evasion of immune surveillance	Degradation of a variety of proteins involved in immune response Impairing cytotoxic T or NK cell-mediated killing Impairing TLR signaling Immune suppression by the recruitment of Treg cells, promoting Th2 cell attraction	[60,61,62,63,64,65,66,67,68,69,70,71,72,73,74,75,76,77,78,79]

## Data Availability

Published data were collected from PubMed.

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
