# Peer review of "Molecular Mechanisms of Kaposi Sarcoma Development"

_cancers, 2022, doi:10.3390/cancers14081869_

Round 1

Reviewer 1 Report

The goal of the review was to provide an overview on the viral and host factors that are involved in the development of KSHV-induced Kaposi’s sarcoma. The review mentions some of the histopathological features of KS and describes some of the viral and host factors that play a role in KSHV-mediated oncogenesis but often the text reads like a summary of reviews instead of discussing original research articles. In many cases the references are reviews and not original papers where the discovery was made. Also, a table and/or a figure would help to better convey the content of the review. There are a number of typos throughout the text, which needs more attention.

  • Line 23 KS Herpes Virus – Kaposi’s sarcoma-associated herpesvirus (KSHV)
  • Line 31 “…degradation of a variety of genes…” genes will be not degraded
  • Line 79-91 Mixed use of KSHV and HHV8. While they refer to the same virus, it would be better to use one name throughout the text. KSHV is usually preferred.
  • Line 93 The most accepted and most widely used correct name of KSHV:
  • Kaposi’s sarcoma-associated herpesvirus and not Kaposi’s Sarcoma Herpes Virus
  • Line 100 What envelope glycoproteins are we talking about here? Viral or cellular?
  • Line 105-107 K12 latent gene(s) was/were left out.
  • Line 122 What are the infectious elements?
  • Line 125 KS is derived from… or KSHV affects…
  • Line 142 KSH- KSHV
  • Line 146 KHSV- KSHV
  • Line 195 What kind of LANA protein repression are we talking about here? LANA is constitutively expressed, even it is induced, during the lytic cycle.
  • Line 224 Ref 51 is a review, it would be better to cite original articles where K1 and K15 were shown in KS samples. Also, in general, it is highly recommended that original articles are cited throughout the text instead of using reviews. This is a review, which should process original articles and not reviewing reviews.
  • No references for the following statement in line 240-242: “Other lytic genes including vPK, viral IL-6 homolog (vIL-240 6/K2), viral interferon regulatory factor 1 (vIRF1/K9), and the membrane proteins K1 and 241 K15 have oncogenic functions.”
  • Line 297 KSHV and not HSHV
  • Line 314 Since KSHV cannot be…

Author Response

Response to  reviewers  comment

Reviewer 1     
The goal of the review was to provide an overview on the viral and host factors that are involved in the development of KSHV-induced Kaposi’s sarcoma. The review mentions some of the histopathological features of KS and describes some of the viral and host factors that play a role in KSHV-mediated oncogenesis but often the text reads like a summary of reviews instead of discussing original research articles. In many cases the references are reviews and not original papers where the discovery was made. Also, a table and/or a figure would help to better convey the content of the review. There are a number of typos throughout the text, which needs more attention.         

We thank the reviewer for his constructive comments. We apologize for the typos and have sought to fix all the errors. We have added some original references instead of review papers and have added a table to better convey the messages of the review.

  • Line 23 KS Herpes Virus – Kaposi’s sarcoma-associated herpesvirus (KSHV)

  • Line 31 “…degradation of a variety of genes…” genes will be not degraded
    We have replaced “genes” with “proteins” in the abstract and in the corresponding sector in the manuscript.

  • Line 79-91 Mixed use of KSHV and HHV8. While they refer to the same virus, it would be better to use one name throughout the text. KSHV is usually preferred.

  • Line 93 The most accepted and most widely used correct name of KSHV:
    Kaposi’s sarcoma-associated herpesvirus and not Kaposi’s Sarcoma Herpes Virus       

  • Line 100 What envelope glycoproteins are we talking about here? Viral or cellular?
    Added.    

  • Line 105-107 K12 latent gene(s) was/were left out.

  • Line 122 What are the infectious elements?
    Corrected: “The late phase ensures the expression of viral structural proteins and the production of the infectious virus”.

  • Line 125 KS is derived from… or KSHV affects…
    Fixed   

  • Line 142 KSH- KSHV
    Fixed

  • Line 146 KHSV- KSHV
    Fixed

  • Line 195 What kind of LANA protein repression are we talking about here? LANA is constitutively expressed, even it is induced, during the lytic cycle.
    Indeed, we have reformulated the sentence: “All these events converge to counteract LANA-mediated repression of lytic gene transcription and the induction of RTA, which transactivates KSHV lytic genes”

  • Line 224 Ref 51 is a review, it would be better to cite original articles where K1 and K15 were shown in KS samples. Also, in general, it is highly recommended that original articles are cited throughout the text instead of using reviews. This is a review, which should process original articles and not reviewing reviews.
    We added references from original articles instead (Ref. 51 and 52). We have also replaced another referenced review with the original article (Ref. 76).

  • No references for the following statement in line 240-242: “Other lytic genes including vPK, viral IL-6 homolog (vIL-240 6/K2), viral interferon regulatory factor 1 (vIRF1/K9), and the membrane proteins K1 and 241 K15 have oncogenic functions.”
    References added       

  • Line 297 KSHV and not HSHV
    Fixed

  • Line 314 Since KSHV cannot be…
    Fixed

Main steps

Main elements

Oncogenic roles

Mechanisms

Refs

Latent phase proteins and RNAs

LANA (ORF73), vFLIP (ORF71), vCyclin (ORF72), Kaposins (ORFK12), RNAs

Cell survival, proliferation, angiogenesis

Telomere maintenance
Increased expression of pro-survival factors
RNA inhibition of apoptosis or immune evasion by targeting various gene transcripts

[41-48]

Lytic activation

Immediate-early (IE) genes
Delayed-early (DE) genes & proteins
Late genes & proteins (K1, K15)

Cell survival & proliferation, viral replication

Production of proangiogenic ( PDGF, VEGF) and pro-inflammatory cytokines (vIL-6), viral oncogenes (vGPCR)
MAPK, PI3K and NFkB pathway activation

[49-54]

Soluble growth factors

Cytokines : VEGF, PDGF, FGF, IL-6, IL-8, tumor necrosis factor alpha, vIL-6
Chemokines

Development & maintenance
Protein synthesis & growth

Autocrine & paracrine pathway signaling
Activation of JAK–STAT, PI3K-AKT, and MAPK–ERK pathways

[55-59]

Immune evasion

Viral proteins: K3, K5, vOX2, RTA, viral interferon regulatory factors (vIRFs)
Viral miRNAs: miR-K12-7
Cytokines: IL6, IL10
Viral chemokines: vCCL1,  vCCL2, vCCL3

Evasion of immune surveillance  

Degradation of a variety of proteins involved in immune response
Impairing cytotoxic T or NK cell-mediated killing
Impairing TLR signaling
Immune suppression by the recruitment of Treg cells, promoting Th2 cell attraction

[60-79]

Table 1. An overview of the mechanisms of Kaposi Sarcoma development

Reviewer 2 Report

The authors provide a comprehensive and extremely well written manuscript highlighting the etiology, incidence, and molecular mechanisms associated with the development of KS. The manuscript is well organized and appropriately referenced. Therapeutic implications are included and highlight existing as well as emerging approaches.

I have no concerns or substantive critique. The manuscript would be of interest to the general readership.

Minor comment:

1. It appears that the word "is" is missing on line 312. "The treatment of KS in the clinic is based on...."

Author Response

Reviewer 2     
The authors provide a comprehensive and extremely well written manuscript highlighting the etiology, incidence, and molecular mechanisms associated with the development of KS. The manuscript is well organized and appropriately referenced. Therapeutic implications are included and highlight existing as well as emerging approaches.

I have no concerns or substantive critique. The manuscript would be of interest to the general readership.

We thank the reviewer for his comments.

Minor comment:

  1. It appears that the word "is" is missing on line 312. "The treatment of KS in the clinic is based on...."
    Corrected

Reviewer 3 Report

The Authors have presented a comprehensive review on molecular mechanism of KS development. Authors tries to cover all the classic information’s and the recent developments in KSHV field from its molecular pathogenesis to therapeutic implications.

I have a few concerns which are as follows:

  • Authors also need to take care of the punctuation and English language throughout the manuscript.
  • Line 72: remove site
  • Line 101: Plz include, KSHV is also known to infect Mesenchymal stem cells (https://doi.org/10.1128/mBio.02109-15)
  • Line 105-107: include K12 (Kaposin) which is also a latency associated gene
  • Line 142: Correct “KSHV has also been reported to infect mesenchymal stem cells”
  • Line 206: Correct “activation of lytic cycles”
  • Authors should also discuss in details about the current therapies, pre-clinical and clinical trials and some of the important in-vitro studies which targets the novel pathways.

For example:

  • Activation of the unfolded protein response by 2-deoxy-D-glucose inhibits Kaposi's sarcoma-associated herpesvirus replication and gene expression (https://doi.org/10.1128/AAC.01126-12).
  • High-dose zidovudine plus valganciclovir for Kaposi sarcoma herpesvirus-associated multicentric Castleman disease: a pilot study of virus-activated cytotoxic therapy. (https://doi.org/10.1182/blood-2010-11-317610)
  • The HIV protease inhibitor nelfinavir inhibits Kaposi's sarcoma-associated herpesvirus replication in vitro. (https://doi.org/10.1128/AAC.01295-10)
  • High levels of LINE-1 transposable elements expressed in Kaposi's sarcoma-associated herpesvirus-related primary effusion lymphoma (https://doi.org/10.1038/s41388-020-01549-9)

I also recommend to authors to include schematic representations of some of the KSHV pathways to keep the interest of the readers.

Author Response

Reviewer 3     
The Authors have presented a comprehensive review on molecular mechanism of KS development. Authors tries to cover all the classic information’s and the recent developments in KSHV field from its molecular pathogenesis to therapeutic implications.
We thank the reviewer for his comments.

I have a few concerns which are as follows:

  • Authors also need to take care of the punctuation and English language throughout the manuscript.
    We apologize for the typos and have sought to correct all the punctuation mistakes.

  • Line 72: remove site
    Done

  • Line 101: Plz include, KSHV is also known to infect Mesenchymal stem cells
    (https://doi.org/10.1128/mBio.02109-15)

  • Line 105-107: include K12 (Kaposin) which is also a latency associated gene

  • Line 142: Correct “KSHV has also been reported to infect mesenchymal stem cells”

  • Line 206: Correct “activation of lytic cycles”

  • Authors should also discuss in details about the current therapies, pre-clinical and clinical trials and some of the important in-vitrostudies which targets the novel pathways.     
    We have added a section in the final “therapeutic implications” part with some data about novel therapies (Ref. 101 and 102).

For example:

  • Activation of the unfolded protein response by 2-deoxy-D-glucose inhibits Kaposi's sarcoma-associated herpesvirus replication and gene expression (https://doi.org/10.1128/AAC.01126-12).
  • High-dose zidovudine plus valganciclovir for Kaposi sarcoma herpesvirus-associated multicentric Castleman disease: a pilot study of virus-activated cytotoxic therapy. (https://doi.org/10.1182/blood-2010-11-317610)
  • The HIV protease inhibitor nelfinavir inhibits Kaposi's sarcoma-associated herpesvirus replication in vitro. (https://doi.org/10.1128/AAC.01295-10)
  • High levels of LINE-1 transposable elements expressed in Kaposi's sarcoma-associated herpesvirus-related primary effusion lymphoma (https://doi.org/10.1038/s41388-020-01549-9)

I also recommend to authors to include schematic representations of some of the KSHV pathways to keep the interest of the readers.       
We have added a table to better convey the messages of the review.    

Round 2

Reviewer 1 Report

My questions and concerns were adequately addressed by the authors.

In Table 1, what are the RNAs in latent phase proteins and RNAs? I assume these would the viral miRNAs. If so, I recommend that the authors say viral miRNAs or v-miRNAs.

Line 162: move "Table 1" from the subheading into the text.

Author Response

Reviewer 1     
My questions and concerns were adequately addressed by the authors.
We thank the author and are glad to have adequately addressed the concerns.

In Table 1, what are the RNAs in latent phase proteins and RNAs? I assume these would the viral miRNAs. If so, I recommend that the authors say viral miRNAs or v-miRNAs.
Viral miRNAS. Corrected.

Line 162: move "Table 1" from the subheading into the text.
Done. We have moved it to the corresponding parts in the text.